# Global SARS-CoV-2 seroprevalence from January 2020 to April 2022: A systematic review and meta-analysis of standardized population-based studies

Isabel Bergeri[1‡]*, Mairead G. Whelan[2‡], Harriet Ware[2‡], Lorenzo Subissi[1‡], Anthony Nardone[1,3], Hannah C. Lewis[1,4], Zihan Li[2,5], Xiaomeng Ma[2,6], Marta Valenciano[1,3], Brianna Cheng[1,7], Lubna Al Ariqi[8], Arash Rashidian[9], Joseph Okeibunor[4], Tasnim Azim[9], Pushpa Wijesinghe[9], Linh-Vi Le[10], Aisling Vaughan[11], Richard Pebody[11], Andrea Vicari[12], Tingting Yan[13], Mercedes Yanes-Lane[14], Christian Cao[2], David A. Clifton[15], Matthew P. Cheng[16], Jesse Papenburg[16], David Buckeridge[16,17], Niklas Bobrovitz[13,18‡], Rahul K. Arora[2,15‡], Maria D. Van Kerkhove[1‡], Unity Studies Collaborator Group[¶]

1 World Health Organization, Geneva, Switzerland, 2 Centre for Health Informatics, Cumming School of Medicine, University of Calgary, Calgary, Alberta, Canada, 3 Epiconcept, Paris, France, 4 World Health Organization, Regional Office for Africa, Brazzaville, Congo, 5 Faculty of Engineering, University of Waterloo, Waterloo, Ontario, Canada, 6 Institute of Health Policy Management and Evaluation, University of Toronto, Toronto, Ontario, Canada, 7 School of Population and Global Health, McGill University, Montreal, Quebec, Canada, 8 World Health Organization, Regional Office for the Eastern Mediterranean, Cairo, Egypt, 9 World Health Organization, Regional Office for South-East Asia, New Delhi, India, 10 World Health Organization, Regional Office for the Western Pacific, Manila, Philippines, 11 World Health Organization Regional Office for Europe, Copenhagen, Denmark, 12 World Health Organization, Regional Office for the Americas (Pan American Health Organization), Washington DC, United States of America, 13 Temerty Faculty of Medicine, University of Toronto, Toronto, Ontario, Canada, 14 COVID-19 Immunity Task Force Secretariat, McGill University, Montreal, Canada, 15 Institute of Biomedical Engineering, University of Oxford, Oxford, United Kingdom, 16 Division of Infectious Diseases and Medical Microbiology, McGill University Health Centre, Montreal, Quebec, Canada, 17 Department of Epidemiology, Biostatistics and Occupational Health, McGill University, Montreal, Quebec, Canada, 18 Department of Critical Care Medicine, University of Calgary, Calgary, Canada

‡ IB, MW, HW, and LS share first authorship on this work. NB, RKA, and MDVK are joint senior authors on this work.
¶ Membership of Unity Studies Collaborator Group is provided in S1 Acknowledgments.
* bergerii@who.int

**Data Availability Statement:** Machine-readable data files for dataset 0, sub-dataset 1, and sub-dataset 2 are available on the following Zenodo link:

## Abstract

### Background

Our understanding of the global scale of Severe Acute Respiratory Syndrome Coronavirus 2 (SARS-CoV-2) infection remains incomplete: Routine surveillance data underestimate infection and cannot infer on population immunity; there is a predominance of asymptomatic infections, and uneven access to diagnostics. We meta-analyzed SARS-CoV-2 seroprevalence studies, standardized to those described in the World Health Organization's Unity protocol (WHO Unity) for general population seroepidemiological studies, to estimate the extent of population infection and seropositivity to the virus 2 years into the pandemic.

https://doi.org/10.5281/zenodo.6915823. Raw early results submitted via the Unity Study Collaborators initiative are available on the following Zenodo link: https://zenodo.org/communities/unity-sero-2021?page=1&size=20. DOIs and citations for the studies in this Zenodo community is included in Table A to C in S1 Materials. The Python code used for our automated estimate prioritization is available on the following GitHub link: https://github.com/serotracker/iit-backend/blob/8059e9b905395de997f28a1a2dff5def795276ad/app/utils/estimate_prioritization/estimate_prioritization.py.

**Funding:** This work was supported by WHO (WHO COVID-19 Solidarity Response Fund, to IB. https://covid19responsefund.org/en/; German Federal Ministry of Health COVID-19 Research and Development Fund, to IB; World Health Organisation funding, to RKA), the Public Health Agency of Canada (Canada's COVID-19 Immunity Task Force through the Public Health Agency of Canada, to RKA, grant number 2021-HQ-000056 https://www.covid19immunitytaskforce.ca/), the Canadian Medical Association (Joule Innovation Fund, to RKA https://joulecma.ca/), and the Robert Koch Institute (funding to RKA https://www.rki.de). IB, LS, AnV, LA, AR, JO, TA, PW, LL, AiV, RP, MVK are employed and receive salaries from WHO (one of the funders of this study), and AN, MV, BC and HCL are WHO consultants. Authors who are members of the SeroTracker Group (led by RKA, including MW, HW, ZL, XM, TY, CC, MYL, JP, MPC, DB, ML, MS, GRD, NI, CZ, SP, HPR, TY, KCN, DK, SAA, ND, CD, NAD, EL, RKI, ASB, ELB, AS, JC) were supported through the aforementioned grants from WHO, Canada's COVID-19 Immunity Task Force through the Public Health Agency of Canada, the Robert Koch Institute, and the Canadian Medical Association Joule Innovation Fund. WHO had a role in the study design, data collection, data analysis, data interpretation, and the writing of the report. No other funders had any such role.

**Competing interests:** I have read the journal policy and the authors of this manuscript have the following competing interests: RKA, MW, HW, ZL, XM, CC, MYL, DB, JP, MPC, ML, MS, GRD, NI, CZ, SP, HPR, TY, KCN, DK, SAA, ND, CD, NAD, EL, RKI, ASB, ELB, AS, JC and NB report grants from Canada's COVID-19 Immunity Task Force through the Public Health Agency of Canada, and the Canadian Medical Association Joule Innovation Fund. RKA, MW, HW, ZL, CC, MYL, NB also report grants from the World Health Organisation and the Robert Koch Institute. RKA reports personal fees

## Methods and findings

We conducted a systematic review and meta-analysis, searching MEDLINE, Embase, Web of Science, preprints, and grey literature for SARS-CoV-2 seroprevalence published between January 1, 2020 and May 20, 2022. The review protocol is registered with PROSPERO (CRD42020183634). We included general population cross-sectional and cohort studies meeting an assay quality threshold (90% sensitivity, 97% specificity; exceptions for humanitarian settings). We excluded studies with an unclear or closed population sample frame. Eligible studies—those aligned with the WHO Unity protocol—were extracted and critically appraised in duplicate, with risk of bias evaluated using a modified Joanna Briggs Institute checklist. We meta-analyzed seroprevalence by country and month, pooling to estimate regional and global seroprevalence over time; compared seroprevalence from infection to confirmed cases to estimate underascertainment; meta-analyzed differences in seroprevalence between demographic subgroups such as age and sex; and identified national factors associated with seroprevalence using meta-regression. We identified 513 full texts reporting 965 distinct seroprevalence studies (41% low- and middle-income countries [LMICs]) sampling 5,346,069 participants between January 2020 and April 2022, including 459 low/moderate risk of bias studies with national/subnational scope in further analysis. By September 2021, global SARS-CoV-2 seroprevalence from infection or vaccination was 59.2%, 95% CI [56.1% to 62.2%]. Overall seroprevalence rose steeply in 2021 due to infection in some regions (e.g., 26.6% [24.6 to 28.8] to 86.7% [84.6% to 88.5%] in Africa in December 2021) and vaccination and infection in others (e.g., 9.6% [8.3% to 11.0%] in June 2020 to 95.9% [92.6% to 97.8%] in December 2021, in European high-income countries [HICs]). After the emergence of Omicron in March 2022, infection-induced seroprevalence rose to 47.9% [41.0% to 54.9%] in Europe HIC and 33.7% [31.6% to 36.0%] in Americas HIC. In 2021 Quarter Three (July to September), median seroprevalence to cumulative incidence ratios ranged from around 2:1 in the Americas and Europe HICs to over 100:1 in Africa (LMICs). Children 0 to 9 years and adults 60+ were at lower risk of seropositivity than adults 20 to 29 ($p < 0.001$ and $p = 0.005$, respectively). In a multivariable model using prevaccination data, stringent public health and social measures were associated with lower seroprevalence ($p = 0.02$). The main limitations of our methodology include that some estimates were driven by certain countries or populations being overrepresented.

## Conclusions

In this study, we observed that global seroprevalence has risen considerably over time and with regional variation; however, over one-third of the global population are seronegative to the SARS-CoV-2 virus. Our estimates of infections based on seroprevalence far exceed reported Coronavirus Disease 2019 (COVID-19) cases. Quality and standardized seroprevalence studies are essential to inform COVID-19 response, particularly in resource-limited regions.

from the Public Health Agency of Canada and the Bill and Melinda Gates Foundation Strategic Investment Fund, as well as equity in Alethea Medical (Outside the submitted work). MPC reports grants from McGill Interdisciplinary Initiative in Infection and Immunity and Canadian Institute of Health Research, and personal fees from GEn1E Lifesciences (Outside the submitted work), nplex biosciences (Outside the submitted work), Kanvas biosciences (Outside the submitted work). JP reports grants from MedImmune (Outside the submitted work) and Sanofi-Pasteur (Outside the submitted work), grants and personal fees from Merck (Outside the submitted work) and AbbVie (Outside the submitted work), and personal fees from AstraZeneca (Outside the submitted work). DB reports grants from the World Health Organization, Canadian Institutes of Health Research, Natural Sciences and Engineering Council of Canada (Outside the submitted work), Institute national d excellence en sante et service sociaux (Outside the submitted work), and personal fees from McGill University Health Centre (Outside the submitted work) and Public Health Agency of Canada (Outside the submitted work). CC reports funding from Sanofi Pasteur (Outside of the submitted work). TY reports working for Health Canada as a part-time Senior Policy Analyst with the COVID-19 Testing and Screening Expert Panel, from Nov 2020-Jun 2021 (Outside of the submitted work). TH reports funding recieved from the United States Centers for Disease Control and Prevention for Columbia University (Outside of the submitted work). Author HCL declares receiving funding as a WHO consultant from WHO Solidarity Response Fund and the German Federal Ministry of Health COVID-19 Research and Development.

**Abbreviations:** AFR, Africa region; AIC, Akaike information criterion; AMR, Americas region; ANOVA, analysis of variance; CLIA, chemiluminescent immunoassay; COVID-19, Coronavirus Disease 2019; ELISA, enzyme-linked immunosorbent assay; EMR, Eastern Mediterranean region; EUR, Europe region; HIC, high-income country; HRP, Humanitarian Response Plan; IFA, immunofluorescence assay; JBI, Joanna Briggs Institute; LFIA, lateral flow immunoassay; LMIC, low- and middle-income country; LOESS, locally estimated scatterplot smoothing; MS, Member States; PHSM, public health and social measure; PRISMA, Preferred Reporting Items for Systematic reviews and Meta-Analyses; SARS-CoV-2, Severe Acute Respiratory Syndrome Coronavirus 2; SEAR, South-East Asia region; VOC, variant of concern; WHO, World Health Organization; WPR, Western Pacific region.

## Author summary

### Why was this study done?

- Serosurveys, or studies capturing information on Severe Acute Respiratory Syndrome Coronavirus 2 (SARS-CoV-2) antibody prevalence, help us understand true rates of infection, vaccination, and indicators of immunity in the population against the virus causing Coronavirus Disease 2019 (COVID-19) and inform public health decision-making.

- Previous global systematic reviews of seroprevalence have highlighted a lack of standardization in study methods and fewer datasets in some regions like low- and middle-income countries.

- Recently, in part via WHO's Unity studies, the quantity and quality of available seroprevalence data has increased, providing the opportunity to understand the true extent of exposure to SARS-CoV-2 and differences by demographic groups, region, and time.

### What did the researchers do and find?

- We meta-analyzed standardized SARS-CoV-2 seroprevalence studies to estimate the proportion of the global population with antibodies against SARS-CoV-2, the virus causing COVID-19.

- By September 2021, global SARS-CoV-2 seroprevalence from infection or vaccination was 59.2%, 95% CI [56.1% to 62.2%].

- Overall seroprevalence rose steeply in 2021 due to infection in some regions (e.g., 26.6% [24.6 to 28.8] to 86.7% [84.6% to 88.5%] in Africa) and vaccination and infection in others (e.g., 9.6% [8.3% to 11.0%] to 95.9% [92.6% to 97.8%] in Europe high-income countries [HICs]). After the emergence of Omicron in March 2022, infection-induced seroprevalence rose to 47.9% [41.0% to 54.9%] in Europe HIC and 33.7% [31.6% to 36.0%] in Americas HIC.

### What do these findings mean?

- Seroprevalence has increased over time, with heterogeneity in dynamics and data robustness between regions.

- Estimates of COVID-19 infections based on seroprevalence data far exceed reported cases.

- It remains important to continue investing in serosurveillance to monitor the COVID-19 pandemic and prepare for future potential emerging viruses.

## Introduction

The Coronavirus Disease 2019 (COVID-19) pandemic, caused by the Severe Acute Respiratory Syndrome Coronavirus 2 (SARS-CoV-2) virus, continues to severely impact population health and healthcare systems. The 604 million cases and 6.5 million deaths reported as of September 7, 2022 [1] underestimate the global burden of this pandemic, particularly in low- and middle-income countries (LMICs) with limited capacity for contact tracing, diagnostic testing, and surveillance [2].

Seroprevalence studies estimate the prevalence of SARS-CoV-2 antibodies. These studies are crucial to understand the true extent of infection overall, by demographic group, and by geographic area, as well as to estimate case underascertainment. As anti-SARS-CoV-2 antibodies are highly predictive of immune protection [3,4], seroprevalence studies are also indicative of population levels of humoral immunity and therefore important to inform scenario modeling, public health planning, and national policies in response to the pandemic. Although seroprevalence provides crucial information on population-level infection dynamics, it is important to note that it does not imply protection against infection and therefore is not an appropriate measure to gauge progress towards herd immunity.

During 2021, many regions have experienced third and fourth waves of SARS-CoV-2 infection [1]; concurrently, some countries have vaccinated most residents, while others remain unable to achieve high vaccine coverage due to challenges with supply and uptake [5]. A new wave of well-conducted seroprevalence studies, including many in LMICs, provides robust estimates of seroprevalence in late 2020 and into 2021 [6–8]. Synthesizing these studies is crucial to understand the shifting global dynamics and true extent of SARS-CoV-2 infection and humoral immunity. While previous global systematic reviews of seroprevalence have been conducted [9–12], these have included only studies that sampled participants in 2020 and pooled seroprevalence across all time points. These meta-analyses also highlight the importance of improved standardization and study quality to enable more robust analysis [9–11].

Estimates of seroprevalence can be difficult to compare systematically across different settings due to variations in design aspects including sampled populations, testing and analytical methods, timing in relation to waves of infection, and study quality and reporting. The World Health Organization's Unity Initiative (henceforth "WHO Unity") aims to help produce harmonized and representative seroprevalence study results in accordance with global equity principles [2]. The WHO Unity population-based, age-stratified seroepidemiological investigation protocol (the SEROPREV protocol) [2] provides a standard study design and laboratory approach to general population seroprevalence studies. WHO Unity and its partners have supported the implementation of SEROPREV by providing financial and technical resources, including a well-performing serologic assay. SEROPREV has been implemented in 74 countries globally and in 51 LMICs as of September 2021 [2]. Synthesizing results aligned with the standard SEROPREV protocol improves study comparability, enabling further analysis of these comparable studies to answer key questions about the progress of the pandemic globally.

This systematic review and meta-analysis synthesized seroprevalence studies worldwide aligned with the SEROPREV protocol, regardless of whether the study received support from WHO. Our objectives were to (i) estimate changes in global and regional seroprevalence over time by WHO region and country income level; (ii) assess the level of undetected infection, by global and regional case ascertainment over time by calculating the ratio of seroprevalence to cumulative incidence of confirmed cases; and (iii) identify factors associated with seropositivity including demographic differences by 10-year age band and sex through meta-analysis, and study design and country-level differences through meta-regression.

## Methods

### Search strategy and study selection

We conducted a systematic review of seroprevalence studies (hereafter "studies") published from January 1, 2020 to May 20, 2022, reported according to the Preferred Reporting Items Systematic review and Meta-Analyses (PRISMA) guideline [13] (File A in S1 Materials). We designed a primary search strategy in consultation with a health sciences librarian in MED-LINE, Embase, Web of Science, and Europe PMC using key terms such as SARS-COV-2, COVID-19, seroprevalence, and serology (full strategy and complete list of search terms in File B in S1 Materials). We attempted to mitigate possible publication bias by including both published articles and unpublished literature such as grey literature, preprints, institutional reports, and media reports. For our secondary search and article capture strategy, we invited submissions to our database through the open-access SeroTracker platform and recommendations from international experts, including literature compiled through the WHO Unity studies initiative. In order to access timely evidence and mitigate challenges with publication delay, we also contacted WHO Unity study collaborators that had not yet made results available to the general public prior to our inclusion dates, to upload their aggregate results to the open access Zenodo research data repository [14]. We accepted these templates up to 20 May 2022 in line with our primary search strategy and screened them according to the same criteria as other sources captured in our primary search. This systematic review and meta-analysis protocol was registered with PROSPERO (CRD42020183634) prior to the conduct of the review (File C in S1 Materials) [15], and searches and extractions conducted per the previously established SeroTracker protocol [16].

Studies were screened, data extracted, and critically appraised in duplicate, with these tasks shared by a team of 13 study authors (listed in Research Contributions section under data curation). We have study team members proficient in English, French, Portuguese, Spanish, and Cyrillic languages—articles in all other languages were translated using Google Translate where possible. Conflicts were resolved by consensus. Inclusion and exclusion criteria aligned with the SEROPREV standardized protocol for general population seroprevalence to minimize possible bias introduced by interstudy heterogeneity and other measures of study quality such as poor assay performance and/or sampling methods (full protocol criteria described in File D and E in S1 Materials). We included cross-sectional or longitudinal cohort studies with the objective of estimating SARS-CoV-2 seroprevalence in the general population. Restricting inclusion to direct population samples such as household surveys would have led to very little data in some regions and times, as these studies are expensive and difficult to conduct. Thus, household and community samples were included, as well as studies where a robust sampling frame was described that approximates to a wider population, such as individuals attending medical services (blood donors, pregnant mothers, primary care attendees) or residual sera taken from patients for a variety of other investigations. Finally, we also included people residing in slum dwellings, and some patient populations in humanitarian settings where the patient population in question was extensive enough to be considered a proxy sample frame (evaluated on a case-by-case basis). Both random and nonrandom (i.e., convenience, sequential, quota) sampling methods were included. Convenience samples must have a clear and defined sampling frame, i.e., studies recruiting volunteers were not included.

Studies had to use serological assays with at least 90% sensitivity and 97% specificity as reported by the manufacturer or study authors through an independent evaluation of the test used (File D in S1 Materials), unless conducted in vulnerable countries as defined in the Global Humanitarian Response Plan (HRP) [17]. We employed exception criteria for HRP countries to ensure representativeness in countries where sometimes lower-performing assays were the

only accessible option due to ongoing humanitarian emergencies. Studies employing dried blood spots as a specimen type must meet this threshold as determined through study author-conducted sensitivity and specificity validation using dried blood spots. Multi-assay testing algorithms were included if the combined sensitivity and specificity met these performance thresholds, using standard formulas for parallel and serial testing [18]. Complex multiple testing strategies (3+ tests used) were reviewed on a case-by-case basis by 2 study members. To accommodate these limitations and ensure study inclusion equity, we included all assay types from HRP countries regardless of their reported performance values, as long as the authors reported an assay that was independently validated from either an in-house evaluation or a WHO-approved head-to-head evaluation [19–21]. Finally, algorithms employing a commercial or author-designed binding assay followed by confirmatory testing by virus neutralization assay were included as they constitute the gold standard in serological evaluation [22].

We excluded studies sampling specific closed populations (such as prisons, care homes, or other single-institution populations), recruiting participants without a clear sampling frame approximating the target population or testing strategy, and studies that excluded people previously diagnosed with or vaccinated against COVID-19 after initial sampling.

## Data extraction, synthesis, and analysis

From each study, we extracted seroprevalence estimates for the overall sample and stratified by age, sex, vaccination status, and timing of specimen collection according to the prespecified protocol. We extracted information on study population, laboratory assay used, any corrections made in estimating seroprevalence (e.g., for population or assay performance), seroprevalence, and denominator. Standardized results uploaded to Zenodo by Unity study collaborators additionally included information on the proportion of asymptomatic seropositive individuals.

Our procedure for the standardized, aggregate early data results submitted by Unity collaborators was to direct study authors to input their results into a formulated standard Excel template designed to match the same data extracted during routine published study extraction. A blank version of the Excel template is available for reference [23]. These templates were uploaded directly into R for analysis in tandem with other studies included in the meta-analysis. Templates were verified by 2 independent reviewers, and we conducted follow-up to complete information with study investigators where needed. In instances where data from early reporting templates we had received were published prior to May 20, 2022, or a partial dataset was previously published, we ensured to de-duplicate these data for analysis. We evaluated cases of duplicated results on a case-by-case basis, prioritizing the authors published version by default but made exceptions where data were more complete, robust, or up-to-date in the submitted templates. Once authors published or preprinted their results, a link to the full source was added to the Zenodo repository.

We critically appraised all studies using a modified version of the Joanna Briggs Institute (JBI) checklist for prevalence studies (File F in S1 Materials) [24]. To assess risk of bias, a decision rule assigned a rating of low, moderate, or high risk of bias based on the specific combination of JBI checklist ratings for that study [25]. This decision rule was developed based on guidance on estimating disease prevalence [26,27] and was validated against overall risk of bias assessments derived manually by 2 independent reviewers for previously collected seroprevalence studies in the SeroTracker database, showing good agreement with manual review (intraclass correlation 0.77, 95% CI 0.74 to 0.80; $n = 2,070$ studies) [25]. Early results from templates were screened and evaluated for risk of bias using the same criteria as studies captured through routine screening processes.

We classified seroprevalence studies by geographical scope (local [i.e., cities, counties], subnational [i.e., provinces or states], or national), sample frame, sampling method, and type of serological assay (File G in S1 Materials). Where an article or source material contains multiple, methodologically distinct serosurveys, we split the article into multiple "studies"—for the purpose of this review, "study" means a distinct estimate. Where multiple summary estimates were available per study, we prioritized estimates based on estimate adjustment, antibody isotypes measured, test type used, and antibody targets measured (full details: File H in S1 Materials). We included multiple estimates per study when broken down by time frame in our analysis over time.

Countries were classified according to WHO region [28], vulnerability via HRP status [17], and World Bank income level [29].

We anchored each estimate to the date halfway between sampling start and end ("sampling midpoint date") to best reflect the time period of the study. To select the most representative and high-quality studies for analysis, we used only subnational or national studies rated low or moderate risk of bias to estimate seroprevalence in the general population over time and identify factors associated with seroprevalence (subdataset 1). We used only national studies rated low or moderate risk of bias to estimate case ascertainment (subdataset 2).

To explore possible causes of heterogeneity among study results, we constructed a Poisson generalized linear mixed-effects model with log link function using the glmer function from the lme4 package in R [30–32]. Independent predictors were defined a priori as WHO region, income group, geographic scope, sample frame, pandemic timing, age, cumulative confirmed cases, and average public health and social measure (PHSM) stringency index [33]. To focus on factors associated with seroprevalence from infection, we included studies where less than 5% of the national population was vaccinated 2 weeks before the sampling midpoint date. We included all a priori predictors in the final model, and to evaluate the importance of each relevant predictor, we compared the Akaike information criterion (AIC) of the final model to all models dropping a single predictor at a time (full details on the model and predictor definitions: File H in S1 Materials).

To estimate seroprevalence in the general population, we first produced monthly country-level estimates by meta-analyzing seroprevalence in each country, grouping studies in a 12-week rolling window considering the infrequent availability of seroprevalence studies in most countries (rma.glmm from R package metafor) [34,35]. We then produced monthly regional estimates by taking weighted averages of country estimates by population, ensuring that country contributions to these estimates are proportional to country population. We stratified these estimates by the expected key sources of heterogeneity among study results: region, income class, and time. We pooled HIC and LMIC together in the Eastern Mediterranean (EMR) and Western Pacific regions (WPR) due to the lower number of studies, and in the Africa (AFR) and South-East Asia regions (SEAR) (the only 2 HICs in these regions had no studies).

We produced monthly global estimates where estimates were available for a majority of regions, calculating global estimates as a population-weighted average of regional estimates to ensure regional representation (full details: File H in S1 Materials). We produced 95% confidence intervals for the mean seroprevalence estimate, reflecting uncertainty in the summary effect size [36], and 95% prediction intervals to give a range for the predicted parameter value in a new study. All numerical results presented are from this stage. To visualize the trend in regional and global estimates over time, we fit a smooth curve to these estimates using nonparametric regression (gam from R package mgcv) [37]. We also summarized the relevant variant genome frequency in each region shared via the GISAID initiative [38].

We also estimated to what extent laboratory confirmed SARS-CoV-2 cases [39] underestimated the full extent of infections based on seroprevalence. For studies that sampled participants in 2021, we used national seroprevalence estimates and vaccination rates [40] to calculate seroprevalence attributable to infection only. In countries administering only vaccines using Spike (S) protein antigens (e.g., mRNA), we calculated the ascertainment ratio using only studies that detected anti-nucleocapsid (N) seroprevalence. In countries administering inactivated vaccines that may generate both anti-S and anti-N responses, we adjusted the reported seroprevalence using a standard formula [41]. We then produced regional and global estimates of seroprevalence using the 2-stage process described above and computed the ratio to the corresponding cumulative incidence of confirmed SARS-CoV-2 cases in the region or globally. We stratified by HIC versus LMIC in all regions.

Aggregated results shared by Unity collaborators reported the proportion of seropositives that were symptomatic at some time point prior to sampling, summarized using the median and interquartile range, and tested for differences in distribution across age and sex groups using analysis of variance (ANOVA).

To quantify population differences in SARS-CoV-2 seroprevalence, we identified studies with seroprevalence estimates for sex and age subgroups. We calculated the ratio in seroprevalence between groups within each study, comparing each age group to adults 20 to 29 and males to females. We then aggregated the ratios across studies using inverse variance-weighted random-effects meta-analysis. The amount of variation attributable to between-study heterogeneity versus within-study variance was quantified using the $I^2$ statistic.

Our main analysis used seroprevalence estimates uncorrected for test characteristics. As a sensitivity analysis, we also produced global and regional estimates adjusting for test characteristics through Bayesian measurement error models, with binomial sensitivity and specificity distributions. The sensitivity and specificity values for correction were prioritized from the WHO SARS-CoV-2 Test Kit Comparative Study conducted at the NRL Australia [19], followed by a multicenter evaluation of 47 commercial SARS-CoV-2 immunoassays by 41 Dutch laboratories [42], and from independent evaluations by study authors where author-designed assays were used.

Data were analyzed using R statistical software version 4.1.2 [32].

# Results

## Study characteristics

We identified 173,430 titles and abstracts in our search spanning from January 1, 2020 to May 20, 2022 (Fig 1). Of these, 5,281 full-text articles were included in full text screening. A total of 513 seroprevalence data sources containing studies aligned with the SEROPREV protocol were identified, 480 published (94%) and 33 aggregated results from collaborators (6%), of which 12 sources were not in one of our main languages and translated via Google Translate. The 513 sources contained a total of 965 unique seroprevalence studies (detailed references and information available in Table A to C in S1 Materials). Over 1,500 full-text articles were excluded due to not containing studies compatible with the SEROPREV protocol; the main reasons for articles' exclusion at this stage was having an incorrect sample frame for this analyses' scope (i.e., we focused on seroprevalence in the general population and therefore excluded 1,073 articles of exclusively healthcare workers, close contacts of confirmed cases, or other specific closed populations) or not meeting our predefined assay quality performance threshold (374 articles).

A total of 52% (100/194) of WHO Member States (MS) and 4 WHO countries, areas, and territories, across all 6 WHO regions, were represented among the seroprevalence studies

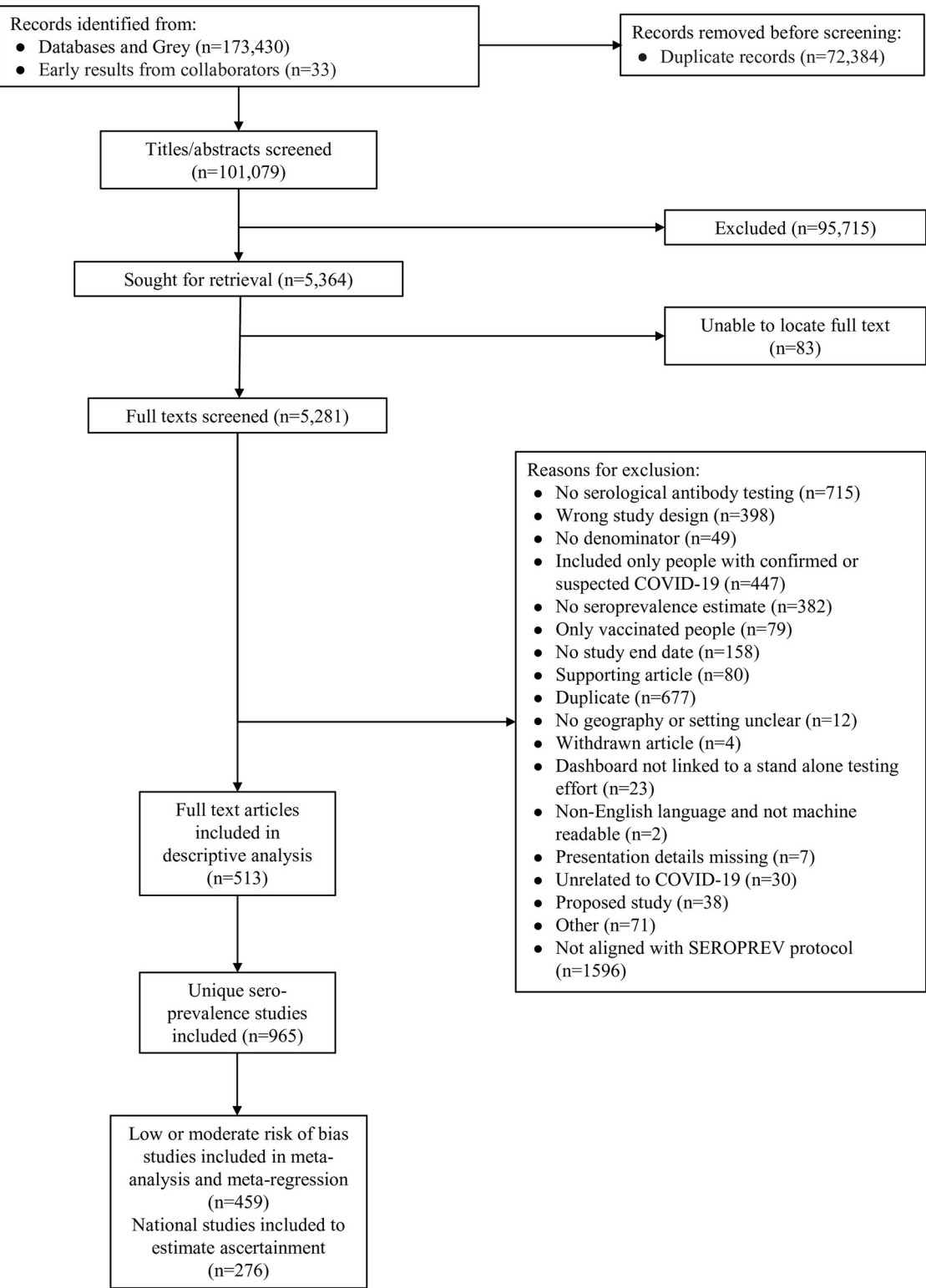

**Fig 1. PRISMA flow diagram of inclusion.** In cases where sources contained multiple primary estimates of seroprevalence (i.e., in nonoverlapping populations, separate methodological seroprevalence studies reported in the same article, etc.), the source (full text) was split into multiple individual studies included in the analysis. For this reason, we report more unique seroprevalence studies than original full-text articles included.

included in the descriptive analysis (Fig A in S1 Materials). Of 47 MS, 23 were represented in AFR; 11 of 21 MS and 1 territory in EMR; 13 of 35 MS and 1 territory in AMR; 39 of 53 MS and 2 territories in EUR; 6 of 11 MS in SEAR; and 8 of 27 MS in WPR (Fig A in S1 Materials). Data from 61 of 134 LMICs and from 36 of 63 vulnerable HRP countries were included. A large proportion of studies included in the descriptive analysis were conducted in LMIC (397/ 965, 41%) and in vulnerable HRP countries (206/965, 21%) (Table 1). Of studies included in the meta-analysis and meta-regression, these proportions were 30% (137/459) and 14% (66/ 459), respectively. Of the 66 (30%) meta-analyzed studies in HRP countries, 20 had test performance values below 90% sensitivity or 97% specificity and were included due to exception criteria.

Among the 965 studies included in the descriptive analysis, 42% (402/965) reported results at a local level, 36% (345/965) at a national level, and 23% (218/965) at a subnational level. The most common sampling frame and method was household and communities (52%, 500/965) and probability sampling (53%, 515/965), respectively. Within household-based samples, only 86/500 studies (17.2%) used convenience sampling. Among the testing strategies used to measure seroprevalence, most studies used ELISA (37%, 360/965) or CLIA assays (37%, 355/965), and few studies used a lateral flow immunoassay (9.4%, 91/965) or multiple assay testing algorithm (7.4%, 72/965). The majority of studies (734/965, 76%) had no vaccination at the sampling midpoint date in the country of the study (Table 1). Very few studies (14/965, 1.5%) in 4 countries (Canada, Japan, United Kingdom, United States of America) sampled participants during 2022.

Most (50%, 483/965) studies were rated moderate risk of bias. A summary of overall risk of bias ratings and breakdown of each risk of bias indicator for all studies is available (Fig B and Table D in S1 Materials, respectively). Subnational and national studies at low or moderate risk were included in the subsequent results.

## Overall and infection-induced seroprevalence by month, region, and income class

We estimated weighted seroprevalence in a series of separate meta-analyses each month and found in September 2021, global seroprevalence from infection or vaccination (overall seroprevalence) was 59.2%, [95% CI 56.1% to 62.2%, 95% prediction interval 51.2% to 66.7%]—an increase since the June 2020 estimate of 7.7% [CI 5.7% to 10.3%, prediction interval 4.2 to 13.8] (Table E in S1 Materials). In September 2021, global seroprevalence attributable to infection was 35.9% [CI 29.5% to 42.7%, prediction interval 22.8% to 51.4%] (Fig 2 and Table E in S1 Materials).

Regional analyses began in January 2020 and ended in February 2021 through March 2022 depending on when seroprevalence studies in each region sampled participants. Overall seroprevalence in February 2021 was 42.7% [37.6% to 48.0%] in EMR (compared to 33.6% [32.8% to 34.4%] in June 2020). In April 2021, overall seroprevalence was 20.0% [18.8% to 21.2%] in AMR LMIC (2.3× since June 2020). In June 2021, overall seroprevalence was 48.7% [47.7% to 49.7%] in EUR LMIC, compared to 22.4% [21.1% to 23.8%] in July 2020. In September 2021, overall seroprevalence was 82.2% [75.9% to 87.2%] in SEAR (8.9× since June 2020). In December 2021, overall seroprevalence was 86.7% [84.6% to 88.5%] in AFR (3.5% [2.9% to 4.2%] in June 2020) and 30.3% [25.3% to 35.9%] in WPR (0.2% [0.1% to 0.4%] in June 2020). Finally, in March 2022, overall seroprevalence was 95.9% [92.3% to 97.8%] in EUR HIC (4.3% [3.4% to 5.5%] in June 2020), and 99.8% [99.7% to 99.9%] in AMR (HIC) (3.6% [2.5% to 5.2%] in June 2020). (Fig 2, middle panel, and Table E in S1 Materials). Infection-induced seroprevalence is reported in Table E in S1 Materials; for example, 47.9% [41.0% to 54.9%] of the population in

**Table 1. Characteristics of included studies, January 2021–May 2022.**

| | ALL STUDIES | LOW AND MODERATE RISK OF BIAS STUDIES; NATIONAL OR SUBNATIONAL SCOPE | LOW AND MODERATE RISK OF BIAS STUDIES; NATIONAL SCOPE ONLY |
|---|---|---|---|
| | Dataset 0 | Subdataset 1 | Subdataset 2 |
| | Used in descriptive analysis | Used to estimate seroprevalence in the general population over time and identify associated factors | Used to estimate case ascertainment[2] |
| **Number of studies** | $N = 965^1$ | $N = 459^1$ | $N = 276^1$ |
| **Study Characteristics:** | | | |
| **Income level** | | | |
| Low-income country | 101 (10%) | 26 (5.7%) | 24 (8.7%) |
| Lower middle-income country | 130 (13%) | 52 (11%) | 25 (9.1%) |
| Upper middle-income country | 166 (17%) | 59 (13%) | 23 (8.3%) |
| High-income country | 568 (59%) | 322 (70%) | 204 (74%) |
| **Vulnerable countries (humanitarian response plan [HRP])** | | | |
| Vulnerable HRP country | 206 (21%) | 66 (14%) | 29 (11%) |
| **WHO region** | | | |
| Africa region (AFR) | 171 (18%) | 47 (10%) | 38 (14%) |
| Americas region (AMR) | 244 (25%) | 117 (25%) | 34 (12%) |
| Eastern Mediterranean region (EMR) | 44 (4.6%) | 21 (4.6%) | 17 (6.2%) |
| Europe region (EUR) | 402 (42%) | 233 (51%) | 172 (62%) |
| South-East Asia region (SEAR) | 65 (6.7%) | 26 (5.7%) | 5 (1.8%) |
| Western Pacific region (WPR) | 39 (4.0%) | 15 (3.3%) | 10 (3.6%) |
| **Geographic scope** | | | |
| Local | 402 (42%) | 0 (0%) | 0 (0%) |
| Subnational | 218 (23%) | 183 (40%) | 0 (0%) |
| National | 345 (36%) | 276 (60%) | 276 (100%) |
| **Study population** | | | |
| Blood donors | 193 (20%) | 86 (19%) | 69 (25%) |
| Residual sera | 197 (20%) | 98 (21%) | 53 (19%) |
| Household and community samples | 500 (52%) | 256 (56%) | 142 (51%) |
| Pregnant or parturient women | 60 (6.2%) | 15 (3.3%) | 11 (4.0%) |
| Persons living in slums | 5 (0.5%) | 1 (0.2%) | 0 (0%) |
| Multiple general populations | 8 (0.8%) | 3 (0.7%) | 1 (0.4%) |
| Representative patient population | 2 (0.2%) | 0 (0%) | 0 (0%) |
| **Sampling method** | | | |
| Convenience sampling[3] | 254 (26%) | 66 (14%) | 29 (11%) |
| Probability sampling | 515 (53%) | 330 (72%) | 198 (72%) |
| Sequential sampling | 178 (18%) | 56 (12%) | 44 (16%) |
| Quota sampling | 18 (1.9%) | 7 (1.5%) | 5 (1.8%) |
| **Test type[4]** | | | |
| CLIA | 355 (37%) | 151 (33%) | 81 (29%) |
| ELISA | 360 (37%) | 175 (38%) | 107 (39%) |
| IFA | 68 (7.0%) | 62 (14%) | 62 (22%) |
| LFIA | 91 (9.4%) | 45 (9.8%) | 15 (5.4%) |
| Luminex | 5 (0.5%) | 2 (0.4%) | 0 (0%) |
| Multiple Assay Testing Algorithm: Binding Assays + Confirmatory Testing with Neutralization Assay | 40 (4.1%) | 8 (1.7%) | 7 (2.5%) |

*(Continued)*

**Table 1.** (Continued)

| | ALL STUDIES | LOW AND MODERATE RISK OF BIAS STUDIES; NATIONAL OR SUBNATIONAL SCOPE | LOW AND MODERATE RISK OF BIAS STUDIES; NATIONAL SCOPE ONLY |
|---|---|---|---|
| | **Dataset 0** | **Subdataset 1** | **Subdataset 2** |
| | **Used in descriptive analysis** | **Used to estimate seroprevalence in the general population over time and identify associated factors** | **Used to estimate case ascertainment[2]** |
| Multiple Assay Testing Algorithm: Other Strategies | 32 (3.3%) | 12 (2.6%) | 3 (1.1%) |
| Other | 8 (0.8%) | 3 (0.7%) | 1 (0.4%) |
| Neutralization | 6 (0.6%) | 1 (0.2%) | 0 (0%) |
| **Overall risk of bias** | | | |
| Low | 183 (19%) | 124 (27%) | 82 (30%) |
| Moderate | 483 (50%) | 335 (73%) | 194 (70%) |
| High | 299 (31%) | 0 (0%) | 0 (0%) |
| **Percent vaccinated at sampling midpoint[5]** | | | |
| 0% | 734 (76%) | 308 (67%) | 158 (57%) |
| Above 0% up to 5% | 70 (7.3%) | 29 (6.3%) | 22 (8.0%) |
| Above 5% up to 10% | 15 (1.6%) | 9 (2.0%) | 7 (2.5%) |
| Above 10% | 146 (15%) | 113 (25%) | 89 (32%) |

[1]n (%). See File G in S1 Materials for definitions.

[2]In the ascertainment analysis, studies conducted in 2021 were adjusted for vaccination. See File H in S1 Materials for details.

[3]Convenience sampling was restricted to studies with a clearly defined sampling frame. See File D in S1 Materials for details.

[4]CLIA, chemiluminescent immunoassay; ELISA, enzyme-linked immunosorbent assay; IFA, immunofluorescence assay; LFIA, lateral flow immunoassay.

[5]Vaccination rates taken from Our World in Data.

EUR HIC (UK studies only) and 33.7% [31.6% to 36.0%] of the population in AMR HIC (Canada studies only) had infection-induced antibodies in March 2022. In the meta-analyses by country with at least 2 studies, 75% (188/250) showed considerable heterogeneity from 75% to 100% [36].

## Ratios of seroprevalence to cumulative incidence

Snapshots of seroprevalence to confirmed case ratios, based on estimated weighted seroprevalence using national studies, are shown in Table 2. Globally, the median ratio was 51.3 infections derived from seroprevalence to 1 reported case (51.3:1) in 2020 Quarter Three, suggesting that around 1.9% of cases were reported, and 10.5:1 in 2021 Quarter Three, suggesting that around 9.5% of cases were reported. In 2020 Quarter Three, the median ratio ranged from 3.4:1 in AMR (HIC) (29.4% of cases reported) to 219.6:1 in EMR (0.5% of cases reported). In 2021 Quarter Three, this ranged from 1.8:1 in AMR (HIC) (55.6% of cases reported) to 176.7:1 in AFR (0.6% of cases reported) (Table 2).

## Subgroup analysis

Asymptomatic seroprevalence by age and sex subgroups for studies reporting subgroups on symptoms are shown in Fig C in S1 Materials. Median asymptomatic prevalence was similar across age groups (ANOVA $p = 0.28$). Median asymptomatic prevalence in males was 64.6% compared to 58.6% in females (ANOVA $p = 0.47$).

Within studies, compared to the reference category of 20 to 29 years old, seroprevalence was significantly lower for children 0 to 9 years (prevalence ratio 0.75, 95% CI [0.67 to 0.84],

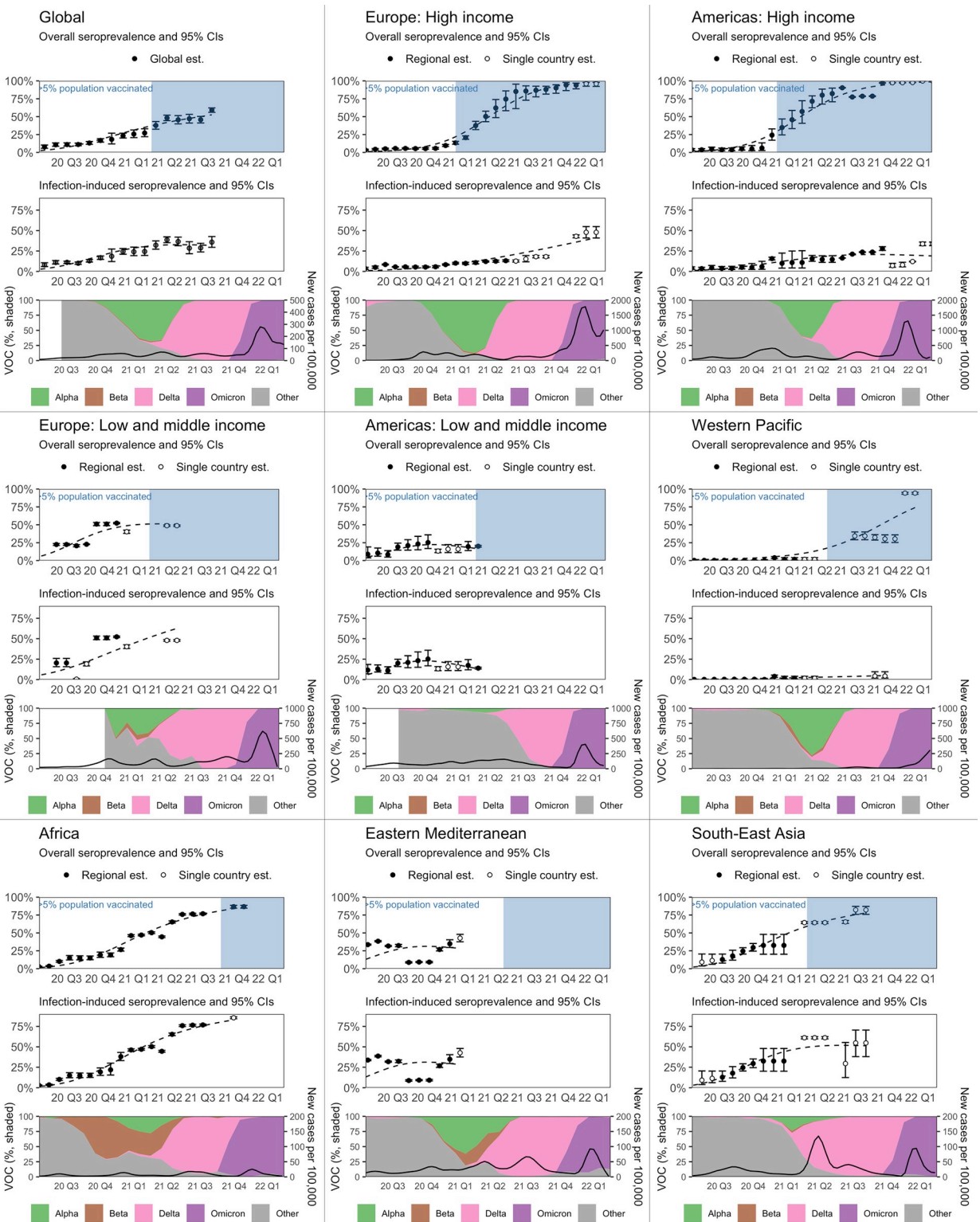

**Fig 2. Estimated seroprevalence globally and by WHO region from January 2020 to March 2022.** The figure contains 9 boxes showing the global analysis and 8 WHO regional analyses. Each box contains the following panels. **Top and middle panel:** We produced weighted point estimates and 95% CIs of overall (top) and infection-induced (middle) seroprevalence by meta-analyzing studies in 12-week rolling windows. To visualize the trend in seroprevalence in each WHO region and globally, we fit a flexible, smooth function of time (dashed line) to the point estimates using nonparametric regression (full details: File H in S1 Materials). Countries included in each region-month estimate are in Table E

in S1 Materials. **Bottom panel, left axis:** Shaded areas represent the relative frequency of major VOCs circulating, based on weekly counts of hCoV-19 genomes submitted to the GISAID we have aggregated by month. Weeks with fewer than 10 total submissions in a given country were excluded from the analysis. **Bottom panel, right axis:** New confirmed cases per 100,000 people, smoothed using local regression (locally estimated scatterplot smoothing: LOESS). Est, estimate; CI, confidence interval; GISAID, Global Initiative on Sharing Avian Influenza Data; VOC, variant of concern; WHO, World Health Organization.

$p < 0.001$) and adults 60+ years (0.87 [0.80 to 0.96], $p = 0.005$). There were no differences between other age groups nor between males and females. (Full results: Fig 3)

## Meta-regression

In the multivariable meta-regression, 329 studies remained after applying our inclusion criteria for prevaccination studies. The full model is reported in Fig 4 (model comparison and diagnostics: Table G in S1 Materials). Subnational studies reported higher seroprevalence estimates compared to national studies (PR 1.27 [1.02 to 1.59], $p = 0.03$). Compared to HIC, higher seroprevalence estimates were reported by low-income (PR 7.33 [3.49 to 15.41], $p < 0.001$), lower middle-income (PR 7.33 [3.49 to 15.41], $p < 0.001$), and upper middle-income countries (PR 3.97 [2.88 to 5.49], $p < 0.001$). Higher cumulative incidence of reported cases was associated with higher seroprevalence (PR 1.39 [1.30 to 1.49], $p < 0.001$), while more stringent PHSM measures up to the sampling midpoint date, continuous from 0 to 10, were associated with lower seroprevalence (PR 0.89 [0.81 to 0.98], $p = 0.02$). Much of the heterogeneity in effect sizes was explained by WHO region, income class, and cumulative confirmed cases. By contrast, sample frame was the least important predictor based on the AIC criterion (Table G in S1 Materials), and compared to studies that sampled households and communities, there were no differences between seroprevalence in studies that sampled blood donors (PR 1.04 [0.77 to 1.40], $p = 0.79$) nor residual sera (PR 1.08 [0.83 to 1.41], $p = 0.55$).

**Table 2. Median estimated seroprevalence to cumulative incidence ratios by WHO region, World Bank income level, and quarter using national studies.**

| WHO region | Income level* | Estimated seroprevalence to case ratios: Median [Range] | | | | |
|---|---|---|---|---|---|---|
| | | 2020 Q3 (July to September) | 2020 Q4 (October to December) | 2021 Q1 (January to March) | 2021 Q2 (April to June) | 2021 Q3 (July to September) |
| Africa (AFR) | Low-middle income (LMIC) | 82.2 [43–104.9] | 152.9 [115.5–156.9] | 154.4 [145.9–211.2] | 185.5 [131.2–226.1] | 176.7 [164.1–189.2] |
| Americas (AMR) | Low-middle income | 22 [6.1–27.5] | 18.3 [18.3–18.3] | NA | NA | NA |
| Americas | High-income (HIC) | 4.6 [2.5–6.5] | 2.6 [2–2.9] | 2.3 [2.2–2.4] | 2 [1.9–2.2] | 1.8 [1.8–1.8] |
| Eastern Mediterranean (EMR) | Low-middle-income | 219.6 [36.9–425.2] | 28.8 [20.8–56.7] | 59.3 [56.8–61.8] | NA | NA |
| Eastern Mediterranean | High-income | 8.8 [8.5–27.2] | 26.2 [25.3–27] | NA | NA | NA |
| Europe (EUR) | Low-middle-income | 84.5 [71.8–97.2] | 44.4 [35.5–53.3] | 25.8 [25.8–25.8] | 11.5 [11.3–11.8] | NA |
| Europe | High-income | 8.1 [5.5–15.4] | 1.9 [1.5–4.3] | 2.1 [1.8–2.1] | 1.6 [1.6–1.8] | 1.9 [1.7–2.1] |
| South-East Asia (SEAR) | Low-middle-income | 30.1 [22.4–37.8] | 40.3 [33.5–48.3] | 37.5 [37.5–37.5] | NA | NA |
| Western Pacific (WPR) | Low-middle-income | 45.4 [29.4–338.9] | 44.3 [38.7–51.6] | 34.5 [31.2–37.9] | NA | NA |
| Western Pacific | High-income | 3.4 [2.9–4] | 3.1 [3–3.2] | NA | NA | NA |
| **Global** | **All** | **51.3 [39.6–62.9]** | **22.7 [20.9–24.5]** | **17.9 [15–20.7]** | **13.5 [13.5–13.5]** | **10.5 [9.3–11.7]** |

NA, national studies not available.

Seroprevalence studies that sampled participants in 2021 were adjusted for antibody target and vaccination rate to calculate seroprevalence attributable to infection (full details: File H in S1 Materials).

*There are no high-income countries in the WHO South-East Asia region; the 2 high-income countries in the WHO Africa region, Mauritius and Seychelles, both have no seroprevalence studies and were hence not included in this analysis.

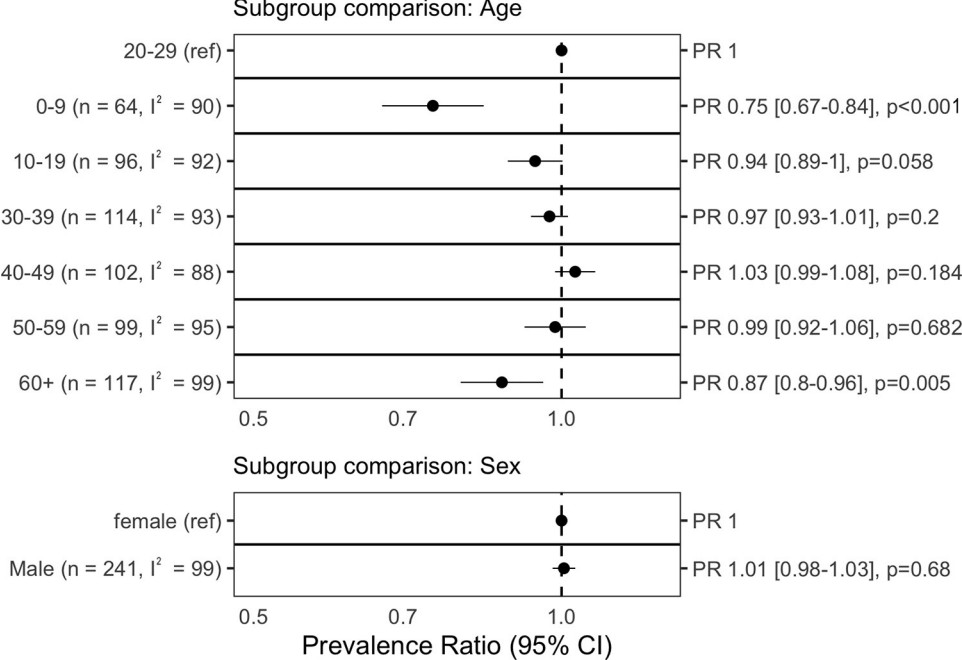

**Fig 3. Meta-analysis of seroprevalence differences by demographic groups.** We calculated the ratio in prevalence between subgroups within each study then aggregated the ratios across studies using inverse variance-weighted random-effects meta-analysis. Each row represents a separate meta-analysis. $I^2$, heterogeneity quantified using the $I^2$ statistic; PR, prevalence ratio.

## Sensitivity analysis

Regional and global estimates of seroprevalence accounting for serological test performance from independent test kit evaluations showed no qualitative differences from the primary results (Table F in S1 Materials). For example, overall global seroprevalence in September 2021 using corrected estimates was 61.5% [56.7% to 66.1%], compared to 59.2% [56.1% to 62.2%] using uncorrected estimates.

## Discussion

We synthesized data from over 800 seroprevalence studies worldwide (43% from LMICs) published up to May 2022 (search dates: January 1, 2020 to May 20, 2022), providing global and regional estimates of SARS-CoV-2 seroprevalence over time with substantial representation of regions with limited available seroprevalence data. We estimate that approximately 59.2% of the global population had antibodies against SARS-CoV-2 in September 2021 (35.9% when excluding vaccination). Global seroprevalence has risen considerably over time, from 7.7% a year before, in June 2020.

Our findings provide evidence of regional and temporal variation in the overall seroprevalence, over 80% in SEAR and AFR in late 2021 and over 90% in AMR HIC and EUR HIC in early 2022. In WPR, there was a paucity of high-quality population-based studies in 2021, and estimated seroprevalence was as low as 30.3% in December 2021, though 1 study in Japan suggests that this has increased to over 90% in February 2022 [43]. Regional variation is driven by differences in the extent of SARS-CoV-2 infection and vaccination. This is exemplified by our monthly timeline of seroprevalence by region, 2020 to 2021, which provides estimates of evolving temporal changes of the global pandemic. We observed increases in seroprevalence

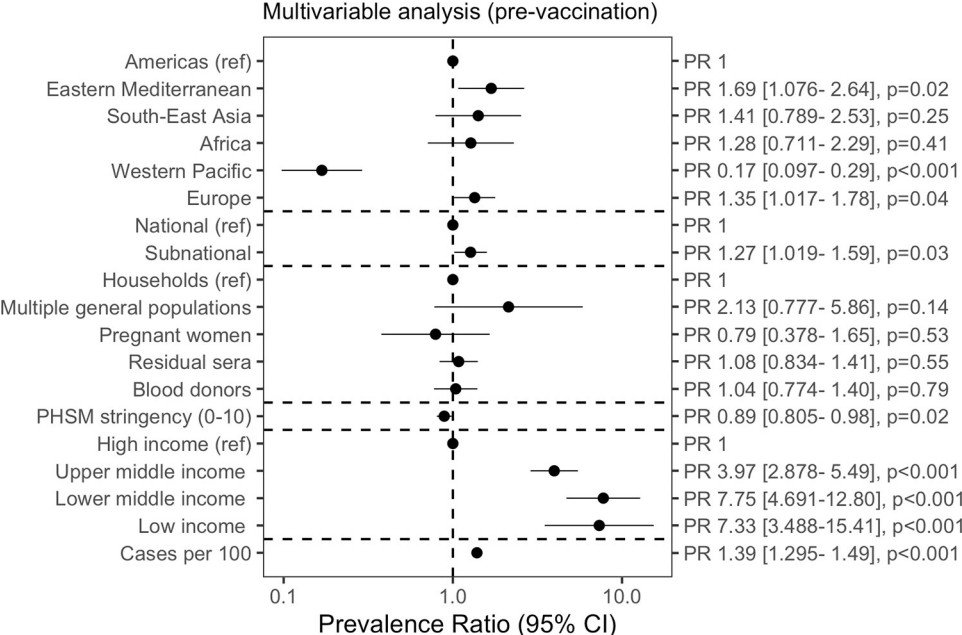

**Fig 4. Meta-regression of seroprevalence (prevaccination) to identify study design and country factors associated with seroprevalence.** We fit a log-Poisson generalized linear mixed-effects model, including studies where less than 5% of the national population was vaccinated 2 weeks before the sampling midpoint date. We performed model comparison using the AIC criterion (Table G in S1 Materials). PHSM data were taken from the London School of Hygiene and Tropical Medicine global dataset. The PHSM index scale ranged from 0 (least stringent) to 10 (most stringent) (see File H in S1 Materials). k = 329; $\chi^2$(95% CI) = 0.74 (0.63–0.87). The marginal R2, or variation between studies explained only by fixed effects, was 62.9%. Multivariable analysis included additional controls for transmission phase and age group not shown in figure. AIC, Akaike information criterion; PHSM, public health and social measure; PR, prevalence ratio.

following the emergence of variants in regions with available data (e.g., 6% (July 2020) to 41% (April 2021) in AFR following the Beta variant and 12% (February 2021) to 75% (August 2021) in SEAR following the Delta variant), demonstrating the substantial number of infections caused by more transmissible variants. In HIC regions, the increases in overall seroprevalence were driven by increased vaccine coverage in early 2021 (e.g., 6% (January 2021) to 95% (August 2021) in AMR HIC and 7% (January 2021) to 72% (August 2021) in EUR HIC), while we also observed increases in infection-induced seroprevalence following the Omicron variant (e.g., 7% (December 2021) to 34% (March 2022) in AMR HIC and 18% (October 2021) to 48% (March 2022) in EUR HIC). Another possibility for regional variation is the potential cross-reactivity of antibodies against *P. falciparum* or other common cold coronaviruses, which has been remarked upon in the literature [44–46], which may impact seroprevalence estimates in areas of Africa or elsewhere with a high incidence of malaria. Our results add global representation and principled estimation of changes in seroprevalence over time as compared to previous evidence syntheses [9,11,12]. These estimates are similar to estimates of true infections by global epidemiological models. For example, our global estimate of seroprevalence attributable to infection (35.9%) is similar to the Institute of Health Metrics and Evaluation cumulative infection incidence estimate of 42.8% on 15 September 2021 [47]. Our analysis provides an orthogonal estimate based solely on seroprevalence data, using a method that has the added value of being easily interpretable and with fewer assumed parameters.

Our results provide evidence of considerable case underascertainment, indicating that many cases of SARS-CoV-2, including subclinical cases, are not captured by surveillance

systems, which in many countries are based on testing of symptomatic patients, have varying sensitivity in their definitions of positive cases, or simply have limited access to testing [48]. There was wide variation in underascertainment (as estimated through seroprevalence–case ratios) in all regions, income groups, and over time, with higher ratios consistently observed in LMICs compared to HICs. Our ratios of seroprevalence to reported cases in late 2020 were comparable to other studies for AMR, EUR, and SEAR [9–12]. Our estimates of seroprevalence to cumulative incidence ratios for AFR, WPR, and EMR are novel, with no other analyses we found having systematically estimated ascertainment through seroprevalence in these regions; moreover, estimates of true infections from epidemiological models suggest that the high levels of underascertainment suggested by this study are plausible [47].

We also provide more granular evidence of significant variation in infection by age by 10-year band. Children aged <10 years, but not children aged 10 to 19, were less likely to be seropositive compared to adults aged 20 to 29 years; similarly, adults aged >60 years, but not those aged 30 to 39, 40 to 49, or 50 to 59, were less likely to be seropositive than adults 20 to 29. These findings add nuance and granularity to differences in seroprevalence by age observed by other studies [10]. Lower seroprevalence in adults 60+ could be explained by immunosenescence that can lead to quicker seroreversion [49] higher mortality and hence a lower proportion of individuals with evidence of past infection, gaps in vaccine access, or more cautious behaviour resulting in fewer infections in this age group. There are several possible explanations for lower seroprevalence in children: milder infections, which are generally associated with lower antibody titers [50]; school closures; and ineligibility for vaccination.

Our multivariable model suggests that higher seroprevalence estimates were reported by low- and lower middle-income countries compared to high-income countries, with the highest seroprevalence in lower middle-income countries (prevaccination). Potential explanations for this result are multifaceted and include weaker health system functionality and performance, lower capacity to isolate, and less stringent use of and ability to effectively implement PHSM. This is also consistent with findings by Rostami and colleagues [11]. Our results suggest that an increase in overall PHSM stringency was associated with lower seroprevalence. This and other work have shown that the use of PHSM was associated with reduced SARS-CoV-2 infections, especially when implemented early and limiting population mobility [51–53]. Our model also found that subnational studies have higher estimates than national studies; one hypothesis for this is that subnational studies are often concentrated on cities or areas with denser populations, which may contribute to increased transmission of the virus. Further research is needed to validate this hypothesis. Finally, our results suggest that blood donors and residual sera studies may be good proxies for the general population, as there was no statistical difference between seroprevalence estimates in these sample frames compared to household and community samples.

In line with the equity principles of the Unity initiative, our dataset had global coverage, including a broad range of LMICs (one-third of studies included in our dataset 1, $n = 177$) and vulnerable HRP countries (14% of included studies). Related other global meta-analyses of seroprevalence had 23% and 35% LMIC coverage, respectively [9,11]. Unity study collaborators shared timely evidence by uploading their aggregated and standardized early results to an open data repository, enabling geographic coverage and reducing publication bias.

## Strengths and limitations

Our regional and global meta-analysis estimates are timely, robust, and geographically diverse with estimates from all WHO regions. The laboratory and epidemiological standardization enabled by the SEROPREV protocol, as well as the analysis of only studies assessed to have low

or moderate risk of bias using a validated risk of bias tool [25], enabled high-quality and comparable data. Despite this effort, there are still methodological differences between the meta-analyzed studies that may reduce their comparability. For example, 14% of studies in our analysis dataset (66/459, 18 of which were household based) were convenience samples, which are less representative than population-based probability samples. To limit this bias, we required Unity-aligned convenience samples to have a clearly defined sample frame (i.e., sampling of volunteers excluded). Our risk of bias evaluation also included subjective review of the demographic breakdown in the study, coverage of subgroup estimates, and author comments on representativeness of the sample, such that the most nonrepresentative studies were rated high risk of bias and excluded from analysis.

A few limitations should be described. First, although we conducted meta-regression to explore heterogeneity of the included studies, there remained some residual heterogeneity that could not be explained quantitatively—likely driven by differences in disease transmission in the different countries and time points that serosurveys were conducted. Second, we did not account for waning of population immunity, so the present work likely underestimates the extent of past infection and case ascertainment. Third, seroprevalence studies are cumulative, meaning that results reflect all COVID-19 countermeasures implemented up to the time of participant sampling, and, thus, we cannot isolate the contributions of particular PHSM. Fourth, while we screened study eligibility based on high assay performance criteria, different serological assays may yield varying results, which should be taken into account when interpreting seroprevalence data. Some argue against combining studies using different assays, because assay performances can vary considerably leading to potential bias in the results. With the moderate seroprevalence values generally observed in our results (roughly 20% to 80%), we expect limited bias to be introduced by the different assays. Nevertheless, we conducted a sensitivity analysis adjusting estimates from individual studies with assay performance whenever available and found that global and regional estimates remained similar. Finally, at certain points in time, our meta-analysis estimates were driven by studies from specific countries—either very populous countries (i.e., SEAR: India, AMR HIC: USA, AMR LMIC: Brazil, WPR: China), or countries in regions with scarce data during the time in question (e.g., EMR: 2 countries in early 2021). We also could not produce global estimates for late 2021/early 2022 due to the delays between when studies conducted their sampling (we extracted from the "sampling midpoint"), and when these results were later published or released within our search dates.

## Implications and next steps

Population-based seroprevalence studies primarily give a reliable estimate of the exposure to infection. In cases where antibodies can be measured quantitatively, it may also be possible to use them to assess the level of protection in a population, although there is currently no consensus on antibody-based correlates of protection for SARS-CoV-2 [4]. While antibodies persist in most infected individuals for up to year (with early evidence pointing at up to 18 months) [54–57], the reinfection risk with the immune-escaping Omicron variant is reported to be much higher than in previous variant of concerns (VOCs) in both vaccinated and previously infected individuals, indicating that the presence of antibodies is less indicative of a level of protection against infection. However, seroprevalence estimates remain indicative of protection against severe disease, as cellular immunity is unlikely to be disrupted even with an immune escaping VOCs.

Seroprevalence studies have been invaluable throughout the COVID-19 pandemic to understand the true extent and dynamics over time of SARS-CoV-2 infection and, to some

extent, immunity. Serosurveillance provides key epidemiologic information that crucially supplements other routine data sources in populations. In populations with reported high vaccine coverage, seroprevalence studies provide a supplement to vaccine coverage data and are an important tool for the evaluation of vaccination programs. In populations with low vaccine coverage, it provides an estimate of cumulative incidence of past SARS-CoV-2 infection (including asymptomatic and mild disease), true case fatality ratio, and avoid many of the limitations of passive disease reporting systems, which can be unreliable due to underdiagnosis and undernotification. Seroprevalence data can be used to compare seropositivity between different groups (age, sex geography, etc.) to identify vulnerable populations and thus inform decisions regarding the implementation of countermeasures such as vaccination programs and PHSM [58]. A key challenge in implementing serosurveillance has been timeliness of study implementation, data analysis, and reporting—as such, it will be important for public health decision-makers to prioritize investment and establish emergency-mode procedures to facilitate timely study implementation early on in future outbreak or variant emergence responses as part of overall surveillance strategy. There is also a need to continue to build national capacity with WHO and other partners to rapidly enable high-quality study implementation and communication of findings in a format friendly to decision-makers. The pandemic persists in large because of inequitable access to countermeasures tools such as vaccines; emphasizing the importance of equitable vaccine deployment globally, the strengthening of health systems and of tailored PHSM to mitigate disease transmission until high population protection is achieved. Globally standardized and quality seroprevalence data continue to be essential to inform health policy decision-making around COVID-19 control measures, particularly in capacity-limited regions with low testing capacity and vaccination rates.

## Conclusions

In conclusion, our results show that seroprevalence has increased considerably over time, particularly from late 2021, due to mainly infection in some regions and vaccination in others. Nevertheless, there is regional variation and over one-third of the global population are seronegative to the SARS-CoV-2 virus. As our understanding of SARS-CoV-2 develops, the role of seroprevalence studies may change including the adaptation of study objectives and methodology to the epidemiological context. Currently, our global estimates of infections based on seroprevalence far exceed reported cases captured by surveillance systems. As we enter the third year of the COVID-19 pandemic, implementation of a global system or network for targeted, multi-pathogen, high-quality, and standardized collaborative serosurveillance [59,60] is a crucial next step to monitor the COVID-19 pandemic and contribute to preparedness for other emerging respiratory pathogens.

## Supporting information

**S1 Materials.** File A. PRISMA checklist. File B. Search strategy. File C. PROSPERO protocol registration. File D. Criteria for WHO Unity alignment of seroepidemiological investigations. File E. SeroTracker inclusion and exclusion criteria. File F. Risk of bias tool breakdown. File G. Screening and extraction methods. File H. Supplementary analysis information. Table A. Studies used in ascertainment analysis. Table B. Additional studies used in meta-analysis and meta-regression. Table C. Additional studies used in descriptive analysis. Table D. Risk of bias breakdown for all studies. Table E. Estimated seroprevalence over time by region and globally (uncorrected for test characteristics). Table F. Estimated seroprevalence over time by region and globally (corrected for test characteristics). Table G. Meta-regression results and model comparison. Fig A. WHO Member States with seroprevalence data identified. Fig B. Risk of

bias assessment for included studies. Fig C. Asymptomatic prevalence in SEROPREV protocol studies by age and sex.
(DOCX)

**S1 Acknowledgements. Unity Studies Collaborator Group.**
(DOCX)

## Acknowledgments

We thank colleagues at partner organizations including WHO/Dubai Logistics Team; Myrna Charles, Kathleen Gallagher, Amen Ben Hamida, Christopher Murrill, Toni Whistler, Venkatachalam Udhayakumar (US Centers for Disease Control and Prevention); Eeva Broberg, Erika Duffell, Maria Keramarou and Pasi Penttinen (European Centre for Disease Prevention and Control), Vincent Richard (Institut Pasteur and the Institut Pasteur International Network); WHO regional offices (Jacob Barnor, Alina Guseinova, Jason M Mwenda, Dmitriy Pereyaslov, Harimahefa Razafimandimby, and WHO HQ (Michael Ryan). We also thank team members from Gabon: Rafiou Adamou, Ayola A Adegnika, Samira Z Assoumou, Rosemary A Audu, Paulin E Ndong, Paulin N Essone, Edgard B Ngoungou. We would especially like to thank all WHO Unity Studies p members, who are also named coauthors of this paper and listed in **S1 Acknowledgments**. These and other collaborators, in all the countries who embarked in this global response effort to COVID-19, are recognized on a dedicated webpage on the Unity and WHO website [https://www.who.int/emergencies/diseases/novel-coronavirus-2019/technical-guidance/early-investigations], as well as all individuals who supported, conducted, or participated in each of the studies supported.

The authors alone are responsible for the views expressed in this publication and they do not necessarily represent the decisions, policy, or views of WHO.

## Author Contributions

**Conceptualization:** Isabel Bergeri, Lorenzo Subissi, Anthony Nardone, Niklas Bobrovitz, Rahul K. Arora, Maria D. Van Kerkhove.

**Data curation:** Mairead G. Whelan, Harriet Ware, Hannah C. Lewis, Zihan Li, Xiaomeng Ma, Tingting Yan, Mercedes Yanes-Lane, Christian Cao, Niklas Bobrovitz.

**Formal analysis:** Harriet Ware, Matthew P. Cheng, Jesse Papenburg, David Buckeridge, Rahul K. Arora.

**Funding acquisition:** Isabel Bergeri, Tingting Yan, Niklas Bobrovitz, Rahul K. Arora, Maria D. Van Kerkhove.

**Investigation:** Isabel Bergeri, Lorenzo Subissi, Anthony Nardone, Hannah C. Lewis, Marta Valenciano, Brianna Cheng, Lubna Al Ariqi, Arash Rashidian, Joseph Okeibunor, Tasnim Azim, Pushpa Wijesinghe, Linh-Vi Le, Aisling Vaughan, Richard Pebody, Andrea Vicari.

**Methodology:** Harriet Ware, David A. Clifton, David Buckeridge, Niklas Bobrovitz, Rahul K. Arora.

**Project administration:** Isabel Bergeri, Mairead G. Whelan, Tingting Yan, Niklas Bobrovitz, Rahul K. Arora, Maria D. Van Kerkhove.

**Resources:** Isabel Bergeri, Lorenzo Subissi, Hannah C. Lewis, Lubna Al Ariqi, Arash Rashidian, Joseph Okeibunor, Tasnim Azim, Pushpa Wijesinghe, Linh-Vi Le, Aisling Vaughan, Richard Pebody, Andrea Vicari.

**Supervision:** Isabel Bergeri, Anthony Nardone, Lubna Al Ariqi, Arash Rashidian, Joseph Okeibunor, Tasnim Azim, Pushpa Wijesinghe, Linh-Vi Le, Aisling Vaughan, Richard Pebody, Andrea Vicari, Tingting Yan, David A. Clifton, Matthew P. Cheng, Jesse Papenburg, David Buckeridge, Niklas Bobrovitz, Rahul K. Arora, Maria D. Van Kerkhove.

**Writing – original draft:** Isabel Bergeri, Mairead G. Whelan, Harriet Ware, Lorenzo Subissi, Anthony Nardone, Hannah C. Lewis, Rahul K. Arora.

**Writing – review & editing:** Isabel Bergeri, Mairead G. Whelan, Harriet Ware, Lorenzo Subissi, Anthony Nardone, Hannah C. Lewis, Zihan Li, Xiaomeng Ma, Marta Valenciano, Brianna Cheng, Lubna Al Ariqi, Arash Rashidian, Joseph Okeibunor, Tasnim Azim, Pushpa Wijesinghe, Linh-Vi Le, Aisling Vaughan, Richard Pebody, Andrea Vicari, Tingting Yan, Mercedes Yanes-Lane, Christian Cao, David A. Clifton, Matthew P. Cheng, Jesse Papenburg, David Buckeridge, Niklas Bobrovitz, Rahul K. Arora, Maria D. Van Kerkhove.

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
