## [Editor Report · Decision Letter 0]

12 Feb 2022

Dear Dr Bergeri, 

Thank you for submitting your manuscript entitled "Global epidemiology of SARS-CoV-2 infection: a systematic review and meta-analysis of standardized population-based seroprevalence studies, Jan 2020-Dec 2021" for consideration by PLOS Medicine.

Your manuscript has now been evaluated by the PLOS Medicine editorial staff and I am writing to let you know that we would like to send your submission out for external peer review.

Please re-submit your manuscript within two working days, i.e. by Feb 16 2022 11:59PM.

Kind regards,

Caitlin Moyer, Ph.D.

Associate Editor

PLOS Medicine

---

## [Decision Letter · Decision Letter 1]

28 Apr 2022

Dear Dr. Bergeri,

Thank you very much for submitting your manuscript "Global epidemiology of SARS-CoV-2 infection: a systematic review and meta-analysis of standardized population-based seroprevalence studies, Jan 2020-Dec 2021" (PMEDICINE-D-22-00442R1) for consideration at PLOS Medicine. 

Your paper was evaluated by a senior editor and discussed among all the editors here. It was also discussed with an academic editor with relevant expertise, and sent to four independent reviewers, including a statistical reviewer. The reviews are appended at the bottom of this email and any accompanying reviewer attachments can be seen via the link below:

[LINK]

In light of these reviews, I am afraid that we will not be able to accept the manuscript for publication in the journal in its current form, but we would like to consider a revised version that addresses the reviewers' and editors' comments. Obviously we cannot make any decision about publication until we have seen the revised manuscript and your response, and we plan to seek re-review by one or more of the reviewers. 

We expect to receive your revised manuscript by May 19 2022 11:59PM. Please email us (plosmedicine@plos.org) if you have any questions or concerns.

We look forward to receiving your revised manuscript. 

Sincerely,

Caitlin Moyer, Ph.D.

Associate Editor

PLOS Medicine

plosmedicine.org

From the academic editor:

1. Seroprevalence studies are sometimes used for other pathogens to gauge progress towards herd immunity. Can the co-authors please elaborate in the text on the utility of serosurveillance for herd immunity for COVID-19 given our knowledge of the virus (and its evolution across different variants)

2. It was good to see WHO recommend serosurveillance in the conclusion. Could the WHO team please provide more detail on what is the public health use of serosurveillance data as part of a national COVID-19 surveillance strategy? Also, what would be the strength of this recommendation, given the delays needed to administer the surveys and the ferocious pace of transmission we've seen in some of these waves that could lead to the survey data being stale by the time they are available to decision-makers? A strong rationale is needed to put the next unit of currency into this rather than improving case surveillance as part of overall laboratory and health system strengthening

3. Can the authors unpack the implications of different assays being used across studies further (including elaborating on why many would argue against combining studies using different assays). The limitation cited in the discussion is helpful but insufficient.

4. The authors mention the underascertainment of cases may explain the difference between seroprevalence rates and estimated infection rates from case surveillance data. This may also be due to some countries using less sensitive definitions than others (see https://wwwnc.cdc.gov/eid/article/28/1/21-1082_article)

5. It was good to see in the covering letter the authors were willing to update the data. July 2021 was long ago (pre Delta wave in many countries and pre Omicron globally). It would be really nice to track the data closer to the end of May 2022 (where Omicron has been able to reach most countries and we would have a more accurate estimate of current global exposure).

Other editorial points:

6. Search: Please update the review to the present time, if possible. Throughout, please clarify the beginning and end dates of the search.

7. Title: Please revise your title according to PLOS Medicine's style. Your title must be nondeclarative and not a question. It should begin with main concept if possible. "Effect of" should be used only if causality can be inferred, i.e., for an RCT. Please place the study design ("A randomized controlled trial," "A retrospective study," "A modelling study," etc.) in the subtitle (ie, after a colon).

8. Line numbers: Please include line numbers, running continuously throughout the text, with the revised version.

9. Abstract: Please structure your abstract using the PLOS Medicine headings (Background, Methods and Findings, Conclusions).

10. Abstract: It may be helpful to provide a sentence with some background for the WHO UNITY protocol.

11. Abstract: Methods and Findings: Please specify the inclusion and exclusion criteria for included studies, and the study designs included.

12. Abstract: Methods and Findings: Please describe the demographic subgroups of interest, such as age and sex, in this sentence: “...meta-analyzed differences in seroprevalence between demographic

subgroups…”

13. Abstract: Methods and Findings: Please mention the study registration as: “The review protocol was registered in PROSPERO under registration number CRD42020183634.” or similar.

14. Abstract: Methods and Findings: Please describe how risk of bias was assessed.

15. Abstract: Methods and Findings: Please report the numbers of participants in the included studies. Please summarize the time periods covered by the included studies.

16. Abstract: Methods and Findings: Please clarify the meaning of “Q2” at first use in the text.

17. Abstract: Methods and Findings: Please present results for the main outcomes, with summary estimates, 95% CIs and p values where applicable. If possible, please mention the number of included studies and participants for each result.

18. Abstract: Methods and Findings: In the last sentence of the Abstract Methods and Findings section, please describe the main limitation(s) of the study's methodology.

19. Abstract: Conclusions: Please address the study implications without overreaching what can be concluded from the data; the phrase "In this study, we observed ..." may be useful.

20. Author summary: At this stage, we ask that you include a short, non-technical Author Summary of your research to make findings accessible to a wide audience that includes both scientists and non-scientists. The Author Summary should immediately follow the Abstract in your revised manuscript. This text is subject to editorial change and should be distinct from the scientific abstract. Please see our author guidelines for more information: https://journals.plos.org/plosmedicine/s/revising-your-manuscript#loc-author-summary

21. Main text: Please use square brackets for in-text citations, placed before the sentence punctuation. Where more than one reference is indicated, please do not include spaces, for example [1].

22. Methods: “We conducted a systematic review of seroprevalence studies (hereafter “studies”) published from 1 January 2020 to 29 October 2021.” Please clarify the dates of the search. In the abstract, and end date of December 30, 2021 is mentioned.

23. Methods: Thank you for the statement regarding the MOOSE guidelines. Please also report your SRMA according to the PRISMA guidelines provided at the EQUATOR site.

http://www.equator-network.org/reporting-guidelines/prisma/

Please provide the completed PRISMA and MOOSE checklists. When completing the checklist, please use section and paragraph numbers, rather than page numbers.

Please add the following statement, or similar, to the Methods: "This study is reported as per the Preferred Reporting Items for Systematic Reviews and Meta-Analyses (PRISMA) guideline (S1 Checklist)."

24. Methods: Please mention any language restrictions for included studies.

25. Methods: Please describe any consideration of publication bias.

26. Methods: Please explicitly mention whether the protocol and analysis plan were established prior to the conduct of the review. Please include the relevant prospectively written document with your revised manuscript as a Supporting Information file to be published alongside your study, and cite it in the Methods section. A legend for this file should be included at the end of your manuscript. Please identify any changes from the protocol in the Methods.

27. Methods: Please mention any methods used to explore possible contributions to high heterogeneity observed in the results.

28. Results: Bottom of page 7: When reporting percentages of studies in descriptive analyses conducted in LMIC/vulnerable HRP, please provide numerators and denominators. Please similarly address the reporting of percentages throughout.

29. Results: For the reporting of comparisons of seroprevalence between age groups and by sex, please report p values. Please also note on page 13 that these results are reported in figure 4, rather than figure 3.

30. Discussion: Please present and organize the Discussion as follows: a short, clear summary of the article's findings; what the study adds to existing research and where and why the results may differ from previous research; strengths and limitations of the study; implications and next steps for research, clinical practice, and/or public policy; one-paragraph conclusion.

31. References: Please use the "Vancouver" style for reference formatting, and see our website for other reference guidelines https://journals.plos.org/plosmedicine/s/submission-guidelines#loc-references

Please check the formatting of each reference, including journal title abbreviations. For example PLOS ONE should be PLoS One. Please remove [Internet] throughout.

32. Figure and Table legend: In the legends, please fully describe all abbreviations used within each figure and table.

33. Figure 2: Please confirm that the appropriate usage rights apply to the use of this map, or please illustrate the included member states using a different map or strategy. PLOS applies the Creative Commons Attribution (CC BY) license to all the works we publish (CC BY 4.0). The maps must be free of logos, copyright symbols / “All rights reserved” text. Please see our guidelines for map images: https://journals.plos.org/plosmedicine/s/figures#loc-maps

34. Figure 3: Please include p values.

35. PRISMA Checklist: Please refer to section and paragraph numbers when completing the checklist (for example: Methods, paragraph 1). Please include the MOOSE Checklist, if it is your intention to do so.

36. S.2.1 Supplementary Methods: As mentioned by reviewer 2, we suggest moving key details of the search strategy and inclusion and exclusion criteria to the main text, especially a clear description of language restrictions and the included sample populations. It would also be helpful to note if the proportions of datasets not yet published/publicly available in the included studies are reported in the results.

37. S.2.4: Please clarify the search dates throughout (here October 29, 2021 is mentioned as the end date).

Comments from the reviewers:

Reviewer #1: Important contribution to our knowledge of the pandemic.

Reviewer #2: See attachment

Michael Dewey

Reviewer #3: The authors present an important manuscript which presents global and national estimates of COVID-19 seroprevalence through a systematic review of existing literature and meta-analysis of seroprevalence data. My major comments are primarily focused on the inclusion criteria used for the meta-analysis.

1. Studies in the meta-analysis include those with non-random sampling methods (convenience, sequential, quota) and/or conducted among study populations that are not necessarily representative of subnational or national populations (i.e. blood donors, pregnant or parturient women, persons living in slums). Unfortunately, they make up a substantial proportion of all studies included in the meta-analysis (Table 1). While it's difficult to assess how much of the data from these studies influence sub-national and national seroprevalence estimates, there is a clear concern for bias. For example: convenience sample - Brooks-Sa 2020. JID. Estimation Without Representation: Early Severe Acute Respiratory Syndrome Coronavirus 2 Seroprevalence Studies and the Path Forward. Please provide adequate details so readers can understand how studies where the study population does not reflect the population of interest and studies with non-random sampling methods contribute to representative estimates.

2. The authors state, "Studies had to use serological assays with at least 90% sensitivity and 97% specificity as reported by the manufacturer or study authors (Supplementary file S2.1), unless conducted in vulnerable countries as defined in the Global Humanitarian Response Plan" and "Data from…33 of 63 vulnerable HRP countries were included." Figure 2 suggests estimates from South America, Africa, and Middle east are primarily consider vulnerable. Without additional information, it's very difficult to assess how this large exception to the inclusion criterion might influence the estimates reported at the national and ultimately at the global level. Were any additional steps taken to account for differences in test accuracy among vulnerable countries?

3. Do the authors have any thoughts on why sub-national studies have higher estimates than national studies (Table S11)?

4. On page 4, the authors state "We also contacted WHO UNITY study collaborators that had not yet made results available to the general public prior to our inclusion dates, to upload their aggregated, standardized results to the Zenodo research data repository". It's unclear whether the analysis includes data external to the systematic literature review. 

5. The seroprevalence to cumulative incidence ratio is a nice ecological approach for exploring case ascertainment. Please clarify how the 9-day criterion used "to account for time from infection to seroconversion" was derived and whether this adjustment was necessary given how much heterogeneity likely exists in sampling durations and mid-points. 

6. There some conflicting information provided regarding time of studies. The methods state studies were published between Jan 2020 and Oct 2021. The abstract and discussion state Jan 2020 to Dec 2021. Please correct or clarify.

Minor comments:

1. Clarify "local" vs. "sub-national"

2. Clarify "articles" vs. "studies" in the main text. This is only mentioned in Figure 1 description.

3. "To focus on factors associated with seroprevalence from infection, we excluded studies where over 5% of the national population was vaccinated two weeks before the sampling midpoint date." Please state how many studies remained after exclusion.

4. S2.1 indicates articles in all other languages were translated using Google Translate where possible. While Google translate is fine for day-to-day translations, using it for a scientific article is uncommon, especially given technical language. How many articles were not in main languages and translated via Google?

5. Explicitly state where VOC data in Figure 3 comes from in the methods

6. Not requested for this manuscript but it would be interesting to see prevalence ratios within smaller age groups among children 0-9 years. It may provide more insight to understanding lower prevalence in young children.

7. On page 14, the authors state "For example, our global estimate of seroprevalence attributable to infection (3.2%) is similar to the Institute of Health Metrics and Evaluation cumulative infection incidence estimate of 37.4% on 31 July 2021." Is this a typo?

8. Although the main purpose of figure 2 is clear, most of the text is not legible due to low resolution.

9. In the supplementary table of contents, please remove the superscript on S1 and correct "Error! Bookmark not defined" 

10. Fix the incomplete sentence in the first paragraph of page 13.

11. "Appendix 6" is referenced on page 14. Please correct.

12. Please include line numbers in future submissions. These are quite helpful for referencing specific items.

Reviewer #4: The authors conducted a systematic review and meta-analysis of SARS-CoV-2 seroprevalence studies after two years in the pandemic to estimate the extent of population infection and remaining susceptibility. They meta-analyzed seroprevalence by country and month, pooling to estimate regional and global seroprevalence over time; estimated the difference between seroprevalence versus infection identified through case reporting. These estimates were also conducted between demographic subgroups and national factors associated with seroprevalence.

This study is a huge undertaking and the authors have been very thorough in designing the study and executing the tasks. The manuscript is well written with clearly stated objectives, well described study design and methods, and the results are clearly displayed. The conclusions are consistent with the findings of the study. Although the authors have accounted for possible limitations of the study, there are some additional ones that the authors may consider:

1. Methods and Discussion: The majority of the seroprevalence studies are community-based. Did the authors investigate the sampling mechanism used in these community-based studies? If the sampling was carried out during sports, music or other mass gathering events, the results may not be as representative as studies in which a random sampling of a community population was carried out. The implications of sampling mechanisms in community-based studies should be discussed 

2. Discussion, P.15 3rd paragraph: The high seroprevalence versus incidence rates in LMICs, especially in Africa: The authors should not take the high rates of seroprevalence for COVID-19 from studies at face value. A number of peer-reviewed publications have shown that false positive results in serological assays may be due to malaria. (e.g. Steinhardt LC, et al. Cross-reactivity of two SARSCoV-2 serological assays in a setting where malaria is endemic. J Clin Microbiol 2021; 59:e00514- 21. https://doi.org/10.1128/JCM.00514-21). The authors should discuss the implications of their findings in light of these publications. 

3. Discussion, P.16 2st paragraph: The authors stated, "In cases where antibodies can be measured quantitatively, such as for SARS-CoV-2, they can also provide correlates with protection against infection." Since a global consensus on the exact targets and antibody concentrations for correlates of protection has not been reached, this statement should be modified. 

Reviewed by Professor Rosanna W Peeling

[LINK]

---

## [Decision Letter · Decision Letter 2]

1 Sep 2022

Dear Dr. Bergeri,

Thank you very much for re-submitting your manuscript "Global SARS-CoV-2 seroprevalence: a systematic review and meta-analysis of standardized population-based studies from Jan 2020-May 2022" (PMEDICINE-D-22-00442R2) for review by PLOS Medicine.

I have discussed the paper with my colleagues and the academic editor and it was also seen again by three reviewers. I am pleased to say that provided the remaining editorial and production issues are dealt with we are planning to accept the paper for publication in the journal.

[LINK]

We look forward to receiving the revised manuscript by Sep 08 2022 11:59PM.   

Sincerely,

Caitlin Moyer, Ph.D.

Associate Editor 

PLOS Medicine

plosmedicine.org

Requests from Editors:

From the Academic Editor:

1. Thank you for your edit to the text of the Discussion at lines 587-589, to clarify that there is no consensus on antibody-based correlates of protection for SARS-CoV-2; however, several other places in the text refer to seroprevalence in helping estimate who is still susceptible. Please edit these accordingly:

-Line 48 "to estimate the extent of population infection and remaining susceptibility."

-Line 79 "however around 40 % of the global population remains susceptible to SARS-CoV-2 infection"

-Line 128 "Synthesizing these studies is crucial to understand the shifting global dynamics and true extent of SARS-CoV-2 infection, humoral immunity, and population susceptibility"

-Lines 602-604 "Seroprevalence data can be used to compare seroprevalence between different groups (e.g. age, sex geography, etc.) to identify susceptible populations and thus inform decisions regarding the implementation of counter measures such as vaccination programs and PHSM [58]."

-Line 620 "Nevertheless, there is regional variation and 40 % of the global population remains susceptible to SARS-CoV-2"

2. Title: Please change the timeframe of the title to reflect the sampling of specimens (e.g. April of 2022) rather than the publication of the articles.

3. Results: Please move the meta-regression from being the first data presented (at line 383) to being between the subgroup analysis and sensitivity analysis (e.g. before line 470).

4. Results: Line 407: In this section, rather than reporting increases as 6.6 fold or 2.2x, etc, please compare differences with a prevalence ratio and 95% CI if possible.

5. Results: Line 428: This section (Ratios of seroprevalence to cumulative incidence) may be difficult to interpret, and it may be easier to understand the completeness of the case surveillance data if percentages (with the denominator representing case burden derived from seroprevalence, and numerator representing the number of reported cases) are used instead.

6. Living systematic review: We would be willing to consider submissions of future versions of the review mapping continued seroprevalence changes. However, publication decisions for any such updates would be dependent upon editorial and peer review.

7. Update to included seroprevalence data: You are welcome to update with more recent seroprevalence results (e.g. anything more recent since May 2022) if you would like to do so.

Other editorial comments:

8. Title: We suggest: “Global SARS-CoV-2 seroprevalence from January 2020 to April 2022: A systematic review and meta-analysis of standardized population-based studies” Please be sure to capitalize the first word of the subtitle.

9. Financial disclosure: Thank you for providing the initials of each author who was supported by each award. Please provide the specific grant numbers and URLs to sponsors’ websites where applicable.

10. Data availability: For the data repository, please provide the link to the most up to date version (https://zenodo.org/record/6915823#.YwOfW3YXZaQ).

“Other relevant data are available in a data repository (doi:10.5281/zenodo.5773152) and/or available from the Zenodo community upon reasonable request.” Please describe the data not available in the repository and please provide details for access/contact information for accessing the data.

Please also mention that the Python code for the automated estimate prioritizaiton is available on GitHub and provide the link.

11. Abstract: Line 69: Please clarify if this should be “in European high-income countries…” or other.

12. Abstract: Please define abbreviations at first use in the text (LMIC, EUR HIC, AMR HIC).

13. Abstract: Methods and Findings: In the last sentence of the Abstract Methods and Findings section, please describe the main limitation(s) of the study's methodology. Please move the sentence beginning “The main limitations of our methodology…” from line 61-63 to the end of the section.

14. Introduction: Line 112-113: Please update the disease burden ( the numbers reported here are out of date).

15. Methods: Line 157-158: Please note where a complete list of the search terms can be found (e.g. in Supporting Information file S2.4).

16. Methods: Line 247: It might be helpful to provide some definitions for geographical scope here.

17. Results: Line 334: Please clarify this to “Google Translate” in this sentence.

18. Results: Line 384-396: Please report p values as p<0.001 where applicable.

19. Results: Line 461: Please report p values as p<0.001 where applicable.

20. Discussion: We suggest that subheadings (e.g. “summary” and “results in context”) could be removed.

21. Discussion: “We synthesized data from over 900 seroprevalence studies worldwide (43% from LMICs) published up to December 2021” Please clarify this sentence as there were a few studies from 2022.

22. Discussion: Please italicize P. falciparum.

23. Page 19: Please remove the contributors, declarations of interests, and data availability sections from the main text of the manuscript. Please ensure all information is completely and accurately entered into the relevant sections of the manuscript submission system.

24. References: Please check the formatting of each reference. Please use the "Vancouver" style for reference formatting, and see our website for other reference guidelines https://journals.plos.org/plosmedicine/s/submission-guidelines#loc-references

25. References: Please double check the information for reference 2.

26. References: Please update reference 6.

27. References: For reference 9 and reference 12 and reference 43, please change the journal abbreviation to PLoS One.

28. Figure 2: Please use p<0.001 where applicable.

29. Figure 3: Please make the headings indicating the WHO regions more visible (e.g. larger text). We suggest referring to “panels” in the legend to make it clear the top, middle, and bottom panels are consistent across all WHO regions presented. In the legend, please fully define GISAID.

30. Figure 4: For the top panel, please indicate “years” on the left axis. Please use I2. Please use p<0.001 where applicable. Please define PR.

31. PRISMA checklist: Please revise the checklist, removing references to line numbers. Please refer to locations within the text using section and paragraph numbers only (e.g. Methods, paragraph 1).

32. Supporting information file S2.2: Rather than inserting a link into the text, please provide a reference to the protocol.

33. Supporting information file S2.4: Please update the search dates.

34. Supporting information: Figure S1: Please confirm that the appropriate usage rights apply to the use of this map. Please see our guidelines for map images: https://journals.plos.org/plosmedicine/s/figures#loc-maps

35. Supporting information Table S11: Please briefly describe the different models in the legend. Please also define AIC and BIC in the legend.

36. Supporting Information S5: References: Please use the "Vancouver" style for reference formatting, and see our website for other reference guidelines https://journals.plos.org/plosmedicine/s/submission-guidelines#loc-references

Comments from Reviewers:

Reviewer #2: The authors have addressed all my extensive comments.

Michael Dewey

Reviewer #3: The authors have addressed all major comments brought up in my earlier review. I recommend a thorough grammatical check prior to publication.

Reviewer #4: The authors have provided comprehensive responses to the editorial and reviewers' comments and suggestions. This work will be useful in providing the rationale and framework for future seroprevalence studies during infectious disease outbreaks nationally, regionally and globally. I have no further comments.

[LINK]

---

## [Editor Report · Decision Letter 3]

12 Sep 2022

Dear Dr Bergeri, 

On behalf of my colleagues and the Academic Editor, Amitabh B. Suthar, I am pleased to inform you that we have agreed to publish your manuscript "Global SARS-CoV-2 seroprevalence from January 2020 to April 2022: a systematic review and meta-analysis of standardized population-based studies" (PMEDICINE-D-22-00442R3) in PLOS Medicine.

Please also address the following editorial points:

-The GitHub link for the Python code (https://github.com/serotracker/iitbackend/

blob/8059e9b905395de997f28a1a2dff5def795276ad/app/utils/estimate_prioritization/e

stimate_prioritization.py) does not seem to work. Please double check the link.

-Title: Please make sure the first word of the subtitle is also capitalized in the manuscript submission system.

-Main text: Throughout text, please place reference numbers within square brackets instead of parentheses, like this [1].

-Reference 49: Please change to “Lancet” as the journal in this reference.

-Figure 3: Please use superscript 2 (e.g. “I^2”) rather than I2 in the figure.

PRESS

Sincerely, 

Caitlin Moyer, Ph.D. 

Associate Editor 

PLOS Medicine